# Laser Operation of Tm: LuAG Single-Crystal Fiber Grown by the Micro-Pulling down Method

**Jian Liu [1,2], Jifei Dong [1], Yinyin Wang [1], Hangqi Yuan [1], Qingsong Song [2], Yanyan Xue [2], Jie Xu [1], Peng Liu [1], Dongzhen Li [1], Kheirreddine Lebbou [3], Zhanxin Wang [1], Yongguang Zhao [1,*], Xiaodong Xu [1,*] and Jun Xu [2,*]**

1   Jiangsu Key Laboratory of Advanced Laser Materials and Devices, School of Physics and Electronic Engineering, Jiangsu Normal University, Xuzhou 221116, China; 1910102@tongji.edu.cn (J.L.); dongjifei@outlook.com (J.D.); tinayinyin24@126.com (Y.W.); yhq20001003@163.com (H.Y.); jiexu7777777@foxmail.com (J.X.); liupeng_tju@126.com (P.L.); dzhl@siom.ac.cn (D.L.); Wanglqwang929@163.com (Z.W.)
2   School of Physics Science and Engineering, Institute for Advanced Study, Tongji University, Shanghai 200092, China; qingsong_song@126.com (Q.S.); xueyanyanf@163.com (Y.X.)
3   Institut Lumière Matière, UMR5306 Université Lyon1-CNRS, Université de Lyon, 69622 Lyon, France; kheirreddine.lebbou@univ-lyon1.fr
*   Correspondence: Yongguangzhao@yeah.net (Y.Z.); xdxu79@jsnu.edu.cn (X.X.); xujun@mail.shcnc.ac.cn (J.X.)

**Abstract:** Single crystals fiber (SCF) of LuAG doped with 2.0 at.% thulium has been grown by using micro-pulling down ($\mu$-PD) technique. A continuous-wave output power of 2.44 W was achieved with a slope efficiency of 11.7% using a 783-nm diode laser as pump source. The beam quality factors $M^2$ were 1.14 and 1.67 in the x and y directions, respectively. This is, to the best of our knowledge, the first report on the Tm:LuAG SCF laser. Using ray tracing analysis, the influence of laser performance by the pump intensity distribution in the SCF was also studied.

**Keywords:** Tm:LuAG; single crystal fiber; mid-infrared laser





## 1. Introduction

Lasers in the 2-μm spectral region attract much attention due to their applications in various fields, such as medical LIDAR system [1] and remote sensing of water vapor in the atmosphere [2]. To date, rare ions $Tm^{3+}$ and/or $Ho^{3+}$ doped materials have been demonstrated promising candidates for the generation of 2 μm laser. In comparison, $Tm^{3+}$ doped laser could be directly pumped by commercially available AlGaAs diodes at ~790 nm and providing a high quantum yield of almost 200% at a high Tm-doping level by taking advantages of the two-for-one cross-relaxation excitation process [3].

$Lu_3Al_5O_{12}$ (LuAG) similar to YAG crystal, has the same isostructural garnet structure and had been widely studied for its excellent thermal conductivity and physical properties [4,5]. In 1995, Barnes et al. first reported the Tm:LuAG laser around 2 μm with a relatively low slope efficiency of 7.3% [6]. In 2015, a maximum output power of 4.42 W at 2021.2 nm was obtained, corresponding to a slope efficiency of 49.5% [7]. In 2020, the Tm:LuAG laser produced a 42.5-μJ pulse energy at a 13.3 kHz repetition rate through the self-Q-switching (SQS) technique [8]. For mode-locking operations, pulse durations of 2 ps and 13.6 ps were respectively generated from Tm:LuAG ceramic [9] and crystal [10] lasers. All these results proved that LuAG is a promising host for the 2-μm laser operation when doped with $Tm^{3+}$ ions.

The micro-pulling down (μ-PD) technique developed from the Czochralski method can grow single-crystal fibers with a diameter around 500 μm and length of several dozen centimeters. In general, we define crystals elongated along one dimensional body of special (normally cylindrical) geometry and diameter from several microns to approximately 1 mm as single-crystal fiber [11]. However, except for a demonstrated laser operation based on a 2-mm diameter Tm:LuAG crystal rod grown using such μ-PD method [12], there is, to the

best of our knowledge, no Tm:LuAG SCF or its based laser operation has been reported so far. Therefore, the laser performance, in particular its dependence on the spatial pumping intensity distribution in such typical SCF, is still unknown.

In this study, Tm:LuAG SCF, with diameter of ~0.9 mm and length of 190 mm, has been grown by using the $\mu$-PD technique. Performance of the Tm:LuAG SCF CW laser pumped by the 783-nm diode laser was investigated. By using ray tracing analysis, the pump intensity distribution in the SCF and thus its impact on laser performance were investigated.

## 2. Experiments

### 2.1. Crystal Growth

To fabricate the Tm:LuAG SCF, the initial raw materials were crackle Tm:LuAG crystals grown by the Czochralski method. The Tm concentration was 2.0 at.% with respect to $Lu^{3+}$ corresponding to the formula $(Lu_{0.98}Tm_{0.02})_3Al_{l5}O_{12}$. The growth experiments were conducted in flowing $N_2$ atmosphere to prevent oxidation of the Ir crucible. The prepared crackle was melt in an iridium crucible, and then, the melt was pulled down continuously through a capillary channel at the bottom of the crucible. An undoped LuAG crystal oriented in <111> direction was used as the seed and pulled down at the rate of 0.5 mm/min. Through a CCD-camera, the meniscus and the diameter of the growing crystal were controlled by controlling the radio-frequency heating power. Figure 1a shows the end view of the SCF with a diameter of 0.9 mm. The crystal with a uniform diameter of 190 mm in length is shown in Figure 1b.

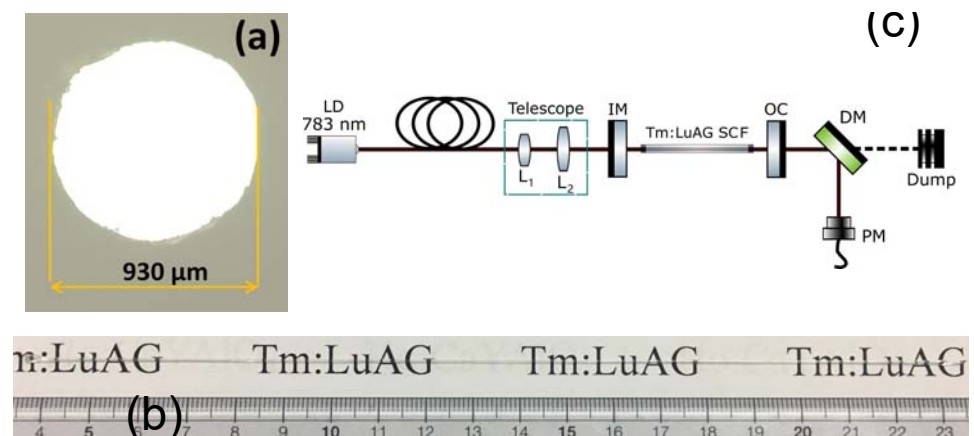

**Figure 1.** End facet of Tm:LuAG SCF (**a**) as-grown Tm:LuAG SCF (**b**) and schematic of the CW Tm:LuAG SCF laser pumped by a 783-nm laser diode (**c**), IM, input mirror; OC, output coupler; DM, dichroic mirror; PM, power meter.

### 2.2. Experimental Setup

The experimental setup of the Tm:LuAG SCF laser is schematically shown in Figure 1c. The pump source was a fiber-coupled (100-$\mu$m core with NA = 0.22) laser diode delivering a maximum power of 25 W at 783 nm with a measured beam quality, i.e., $M^2$-factor of 34. The pump beam was focused into the crystal by a coupling lens system, where L1 with a focal length of 25.4 mm was used to collimate the pump light, and L2 with focal lengths of 30, 50, and 75 mm was used to focus the pump beam into the Tm:LuAG SCF with beam spot diameters of ~116, 196, and 294 $\mu$m. The input mirror (IM) has a high transmission of about 99% at pump wavelength and high reflection of more than 99.9% at the lasing wavelengths. Several output couplers (OCs) with transmission of 3%, 5%, 10%, and 14% were used to explore the best laser performance. A 20-mm-long Tm:LuAG SCF cutting from the fabricated entire SCF was used as the sample for the laser operation. To avoid the Fresnel reflection, both its end faces were polished and anti-reflection coated. We have directly measured the single-pass propagation loss using a He–Ne laser and high-sensitive power meter, giving a loss of 0.048 $cm^{-1}$. The SCF was mounted in a home-made

aluminum module and directly water-cooled to 15 °C. A dichroic mirror (DM) was used as a beam splitter for pump and laser beams.

## 3. Results and Discussion

At first, laser performances with different pump-guiding conditions were investigated. In this case, the focused beam diameters in the SCF were 116 and 196 μm, exhibiting depths of focus of ~0.8 and ~2.4 mm, respectively. The large divergence angle (0.15 and 0.05) made it easy to realize the pump-guiding. Using ray tracing analysis [13], the stimulated spatial distribution of pump light (with L2, *f* = 30) in the SCF is shown in Figure 2a. The pump-guiding mode can be clearly seen. For the beam diameter of 196 μm, the maximum output power was 1.24 W with a slope efficiency of 6.7% at an absorbed pump power of 19 W, higher than that of 0.68 W and 4.6% obtained with the beam diameter of 116 μm. For the latter case, the output power rose slowly with the slope efficiency of 1.5% when the absorbed pump power was beyond 15 W. It seems that the intense pump-guiding gave a worse laser performance in our case with the 20-mm-long Tm:LuAG SCF, which failed to meet our expectations on pump guiding as that in the traditional glass fiber lasers.

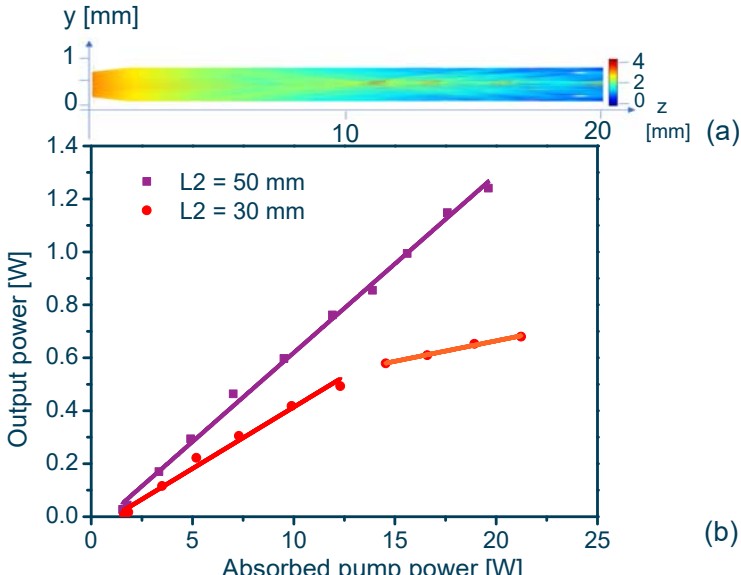

**Figure 2.** Simulated pump intensity distribution in the SCF with L2 (*f* = 30 mm) by using ray tracing analysis (**a**), and output power of Tm:LuAG SCF in different pump configurations (**b**).

Thus, mode-matching may still be a critical factor for power scaling in such relatively short SCF, and thereafter, pumping configuration with a spot diameter of 294 μm was employed, which in principle has a better mode-matching with the fundamental laser mode (~300 μm beam diameter) along the SCF. In this case, the focused pump beam had a depth of focus of ~6 mm, and thus the position of the focal point in the SCF became a critical parameter. Figure 3a shows the stimulated spatial distribution of pump light in the SCF. In the present work, L = 1, 4, 7 mm (distance between the input end facet and focal point), with free propagation of 10, 13, 16 mm in the SCF, was performed for the CW regime. The laser performances for the three pumping configurations were investigated with a physical cavity length of ~2 cm and a 10% OC, as shown in Figure 3b. The maximum output powers for the three cases were 2.16, 2.44, and 1.91 W, corresponding to the slope efficiency of 10.2%, 11.7%, and 8.19%. The case of L = 4 mm has the best laser performance and lowest laser threshold, which indicates a trade-off between mode-matching and pump guiding is essential for the present Tm:LuAG SCF laser.

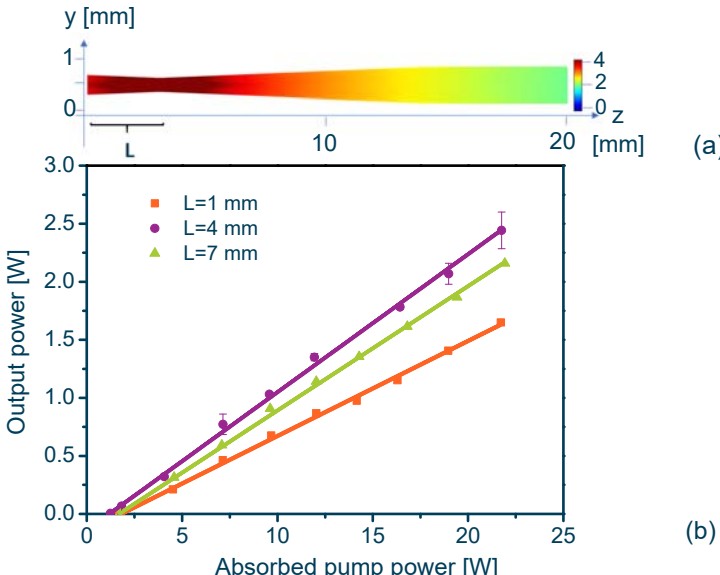

**Figure 3.** Simulated pump intensity distribution in the SCF by using ray tracing analysis (**a**), and output power of Tm:LuAG SCF for the focal points at different positions (**b**), L: distance between the input end facet and focal point.

The CW laser operation for different OCs with a pump beam diameter of 294 μm for the L = 4 mm was thereafter studied in detail. Figure 4 shows the output power with respect to the absorbed pump power. For various OCs, the output power reaches the highest 2.44 W with the 10% OC, and decreased sharply to 1.18 W as the transmittance of the output coupler increased to 14%. The slope efficiency measured this time is much lower than in a previous report about Tm:YAG lasers using a thick rod [14] (50%), and also lower than the Tm:YAG crystal [12] (39.1%) grown by using the μ-PD technique in which the 790-nm LD was used as pump power. The relatively low concentration of $Tm^{3+}$ doped may be responsible for the low slope efficiency. We know that high Tm concentration doped was important to realize two-for-one cross-relaxation. Normally, materials with 4.0 at.% Tm doped were used for laser operation [9,10], while just 2.0 at.% Tm doped was used in our experiments. The detailed data are also shown in Table 1.

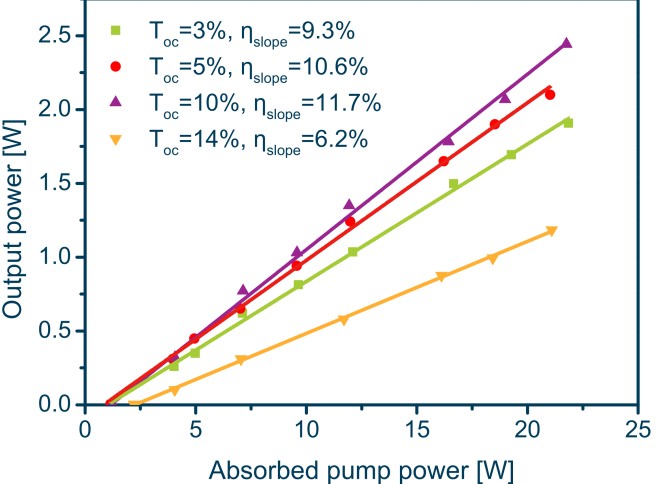

**Figure 4.** The output power of the Tm:LuAG laser as a function of absorbed pump power with different OCs.

**Table 1.** Maximum output power, slope efficiency, and laser threshold for the Tm:LuAG SCF.

| OCs | Threshold (W) | Maximum Output Power (W) | Slope Efficiency (%) |
|---|---|---|---|
| 3% | 1.13 | 1.90 | 9.3 |
| 5% | 1.15 | 2.10 | 10.6 |
| 10% | 1.23 | 2.44 | 11.7 |
| 14% | 2.20 | 1.18 | 6.2 |

Figure 5 shows the laser spectrum of the Tm:LuAG SCF with different OCs. The peak wavelength changed from 2025 nm to 1972 nm. The wavelength red-shift with decrease of the OCs is caused by the enhanced reabsorption effect with lower OC transmission, where a higher population inversion is needed, resulting in a drift of gain spectrum [15]. It is obvious that the multiple wavelengths occurred and the peak wavelength blue-shift rapidly when the OCs increased from 10% to 14%. The change of gain cross sections may be responsible for this phenomenon [16]. Unfortunately, the gain cross section of the Tm:LuAG SCF cannot be depicted for the uneven concentration around the end facet of Tm:LuAG SCF grown by the $\mu$-PD technique [17].

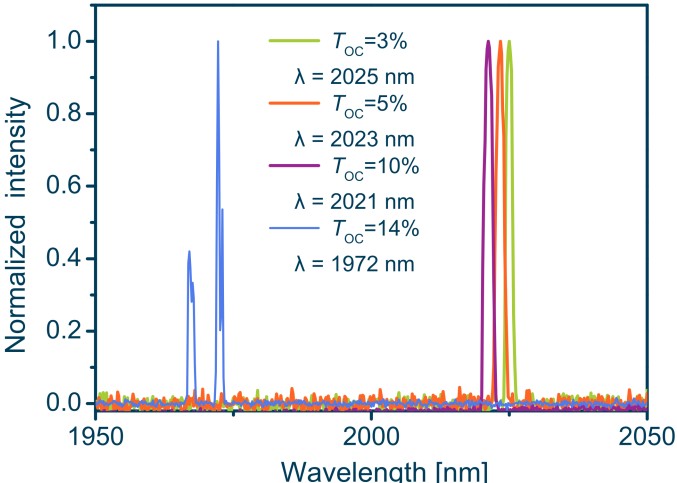

**Figure 5.** The laser spectrum of Tm:LuAG SCF with different OCs.

The beam profile was also measured at an output power of 2 W by using two plano-convex spherical lens (L3, $f$ = 150 mm, L4, $f$ = 100 mm) and a beam profiler (WinCamD-IR-BB, Dataray Inc). The $M^2$ at two directions were measured to be 1.67 and 1.10 in the $x$- and $y$-directions, as shown in Figure 6. The relatively larger $M^2$ value along the $x$-axis is caused by the slight misalignment between the pump and oscillation in this axis, which would introduce more high-order mode. The transverse-mode also can be observed from the 2D beam profile. This phenomenon can be understood since our cavity was plano–plano and no extra transverse-mode-limiting optical elements were applied [16].

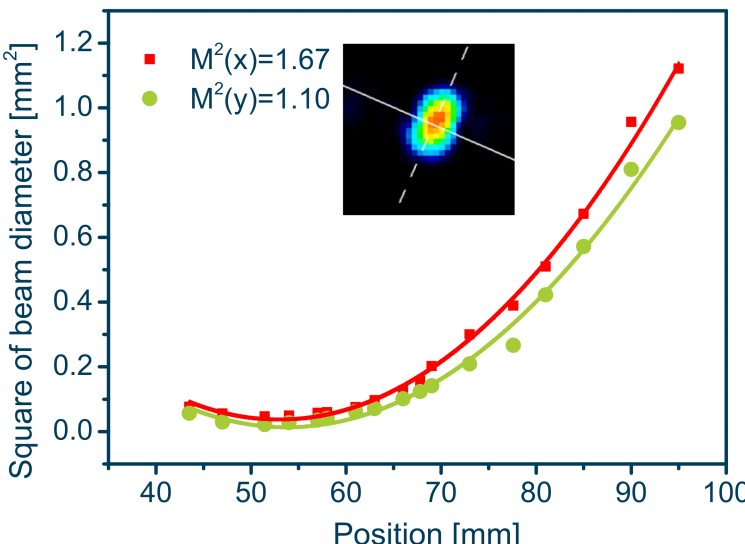

**Figure 6.** The beam quality factor measurement of the Tm:LuAG SCF at 2 W, the inset shows the 2D beam profile.

### 4. Conclusions

　　Tm:LuAG SCF with a diameter of ~0.9 mm and 190 mm in length has been grown by using the $\mu$-PD technique. To the best of our knowledge, the laser performance of Tm:LuAG with a diameter below 1 mm is reported for the first time. The laser operation with different pump-guiding configurations was investigated. The results showed that the pure pump-guiding mode was not a good choice for our Tm:LuAG SCF. Then, the correlation between the free propagation distance and pump-guiding mode was experimentally investigated. The highest output power of 2.44 W with slope efficiency of 11.7% was achieved. Additionally, the beam quality factor was calculated to be 1.10 and 1.67 at two directions. The work illustrates that the trade-off between mode-matching and pump guiding is essential for the present Tm:LuAG SCF laser. Further work will focus on the enhancement of the slope efficiency by optimizing the cavity design and the improvement of crystalline quality with higher $Tm^{3+}$ concentration.

**Author Contributions:** Methodology, Y.Z.; formal analysis, J.D., Y.W., H.Y. and J.X. (Jun Xu); investigation, Q.S., Y.X., J.X. (Jie Xu), P.L. and D.L.; validation, Y.Z.; Data curation, Z.W.; funding acquisition, Y.Z., X.X. and J.X. (Jun Xu); supervision, J.X. (Jun Xu); writing—original draft preparation, J.L.; resources, Y.Z.; writing—review and editing, Y.Z., X.X., J.X. (Jun Xu) and K.L. All authors have read and agreed to the published version of the manuscript.

**Funding:** National Natural Science Foundation of China (62075090, 52032009, 61621001, 61975071) and Natural Science Foundation of Jiangsu Province, China (SBK2019030177).

**Conflicts of Interest:** The authors declare no conflict of interest.

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
