# Peer review of "Laser Operation of Tm: LuAG Single-Crystal Fiber Grown by the Micro-Pulling down Method"

_crystals, doi:10.3390/cryst11080898_

Round 1

Reviewer 1 Report

the authors describe in this paper the development of a fiber obtained by means of the micro-pd technique for the study of a laser operating in the region of two microns. in particular the fiber is monocrystalline LuAG doped with 2% Tm3+.
The article is interesting but there are some aspects that I would like to be deepened.
The authors claim to have obtained a uniform diameter (point 65) on a length of 190 mm. It would be most appropriate to report a measure of this uniformity.
The authors write that the drawing speed is 0.5 mm/minute. Has the LuAG crystal been checked for scattering inside the crystal for the formation of microbubbles? Since the material is cubic, has the presence of optical stress been checked ?
For this kind of material a result of this type is typical. This could also be related to the low laser efficiency obtained in the publication.
Since the laser efficiency has been obtained as a function of OC transmission, it is appropriate to estimate the losses of the medium (for example by finday-clay). In this way it is possible to give an estimate of the optical quality of the fiber. A clue is also the output beam asymmetry.
In conclusion before this manuscript can be published it is necessary to complete it to further investigate the points raised.

Author Response

the authors describe in this paper the development of a fiber obtained by means of the micro-pd technique for the study of a laser operating in the region of two microns. in particular the fiber is monocrystalline LuAG doped with 2% Tm3+.
The article is interesting but there are some aspects that I would like to be deepened.
The authors claim to have obtained a uniform diameter (point 65) on a length of 190 mm. It would be most appropriate to report a measure of this uniformity.

Response: Thank you for your comments.

The diameter of the fiber was measured with vernier calipers in several positions. The diameters of both ends of the sample were also measured with a microscope. After consideration, we decide to change the description to “The 190 mm long crystal is shown in Figure 1(b).”

The authors write that the drawing speed is 0.5 mm/minute. Has the LuAG crystal been checked for scattering inside the crystal for the formation of microbubbles? Since the material is cubic, has the presence of optical stress been checked ?
For this kind of material a result of this type is typical. This could also be related to the low laser efficiency obtained in the publication.

Response: Thank you for your comments.

No bubbles were seen by the transmission electron microscope. The optical stress has not been checked. But high laser efficiency was achieved for the garnet (Ho:YAG SCF [1]) grown using the same method in our team work. So we believe that the low concentration of doping is the dominant reason of the low laser efficiency, not the quality of the crystal.

[1] Zhao, Y. G.; Wang, L.; Chen, W. D.; Wang, J. L.; Song, Q. S.; Xu, X. D.; Liu, Y.; Shen, D. Y.; Xu, J.; Mateos, X.; Loiko, P.; Wang, Z. P.; Xu, X. G.; Griebner, U.; and Petrov, V. 35 W continuous-wave Ho:YAG single-crystal fiber laser. High Power Laser Sci. Eng. 2020, 8. e25.

Since the laser efficiency has been obtained as a function of OC transmission, it is appropriate to estimate the losses of the medium (for example by finday-clay). In this way it is possible to give an estimate of the optical quality of the fiber. A clue is also the output beam asymmetry.

Response: Yes, the propagation loss can be estimated from the dependence of laser efficiency on the OC transmission. Basically at least five OC transmissions are required to precisely estimate the loss. However, in the present work we have only four different OCs for the lasing test, so we did not perform the loss evaluation. Nevertheless, we have directly measured the single-pass propagation loss using a He-Ne laser and high-sensitive power meter, giving a loss of 0.05 cm-1, similar to the report of Ho:YAG SCF.

The corresponding content has been added in the revision and marked in red fonts.

Reviewer 2 Report

The data presented in the figures lack any error bars. I suggest the authors to provide such piece of info.

Author Response

Thank you for your comments, and we have added the error bars for the Figure 3 for the L=4 mm, which achieved the maximum output power. As can been seen, the output power in our experiments is stable after optimizing the cavity.

Reviewer 3 Report

The presented paper describes newly preparated single crystal fiber laser based on thulium–doped LuAg crystal grown by the micro-pulling down method. The Authors present several characteristics of this laser, especially slope characteristics dependent on various pump configurations and on cavity parameters, spectrum of laser action at four wavelengths and characteristic of laser beam quality.  The obtained crystal in the form of “fiber” and placed in the proper cavity revealed laser action quite easily, although its parameters were not as good as those of thulium lasers known so far.

The paper is rather well written, though, to me, English still needs some assistance. The conclusions are consistent with observations and convincing. Hence I do not have any critical remarks regarding this content of the paper which concerns physics. However, I have just few comments concerning some technical and methodological aspects of the presented paper.

First, in Fig. 1 the diameter of the end facet of the “fiber” sample is described as 930 nm, whereas in the text the Authors say about ~ 0.9 mm. If so, in Fig. 1 it should be 930 micrometers.

The last phrase just before this figure can be confusing, I would propose: “The 190 mm long crystal with an uniform diameter is shown in Figure 1(b).”

The next my comment concerns the frequently used: “mode”, “mode-matching” etc. I think, the Authors should comment in the text of which kind of mode they say. This is why I write “fiber” (in quote) in terms of this paper. To deal with optical fiber with developed mode structure we should have fiber of diameter somewhat comparable with propagated wavelength. The sample of ~ 1 mm resembles rod rather than fiber and behaves rather like bulk sample.  Unless the “mode” concerns the cavity modes, but then it should be clearly written in the text.

On page 3 in Fig. 2 the description of axis “x” is missing. On the same page, row 9 from down: it should be “simulated” instead of “stimulated”, and in row 7 from down: “free propagation” instead of “freely propagation”.

On p. 4 rows 2-4 from down I would propose: “..high Tm concentration doping was important to realize two-for-one cross-relaxation. Normally, materials doped with 4.0 at.% were used for…”

P. 5 row 5: it should be: “It is obvious that the multiple wavelength occurred and the peak wavelength blue-shifted rapidly when..”.

P. 5 row 2 from down: I would prefer “plane-parallel” instead of “plano-plano”.

P. 6 row 3 from down (in Conclusion): should be: “…for the presented Tm:LuAG SCF laser.”.

In conclusion I think that the paper could be published in the Crystals, but after taking my comments into account.

Author Response

The presented paper describes newly preparated single crystal fiber laser based on thulium–doped LuAg crystal grown by the micro-pulling down method. The Authors present several characteristics of this laser, especially slope characteristics dependent on various pump configurations and on cavity parameters, spectrum of laser action at four wavelengths and characteristic of laser beam quality. The obtained crystal in the form of “fiber” and placed in the proper cavity revealed laser action quite easily, although its parameters were not as good as those of thulium lasers known so far.

The paper is rather well written, though, to me, English still needs some assistance. The conclusions are consistent with observations and convincing. Hence I do not have any critical remarks regarding this content of the paper which concerns physics. However, I have just few comments concerning some technical and methodological aspects of the presented paper.

First, in Fig. 1 the diameter of the end facet of the “fiber” sample is described as 930 nm, whereas in the text the Authors say about ~ 0.9 mm. If so, in Fig. 1 it should be 930 micrometers.

Response: Thank you for your comments.

And we are sorry for our negligence, and we have modified to 930 μm Figure.1.

The last phrase just before this figure can be confusing, I would propose: “The 190 mm long crystal with an uniform diameter is shown in Figure 1(b).”

Response: Thank you for your comments. We have followed your suggestion.

The next my comment concerns the frequently used: “mode”, “mode-matching” etc. I think, the Authors should comment in the text of which kind of mode they say. This is why I write “fiber” (in quote) in terms of this paper. To deal with optical fiber with developed mode structure we should have fiber of diameter somewhat comparable with propagated wavelength. The sample of ~ 1 mm resembles rod rather than fiber and behaves rather like bulk sample. Unless the “mode” concerns the cavity modes, but then it should be clearly written in the text.

Response: Thank you for your comments. What we inferred “mode” in the text concerns the cavity mode. And we comment in the text. “So mode-matching of the pump beam and the laser maybe still a critical factor for power scaling in such relative short SCF, and thereafter pumping configuration with a spot diameter of 294 µm was employed which in principle has a better mode-matching with the fundamental laser mode (~ 300 µm beam diameter) along the SCF.

Then the readers could better understand what we inferred in the next text.

On page 3 in Fig. 2 the description of axis “x” is missing. On the same page, row 9 from down: it should be “simulated” instead of “stimulated”, and in row 7 from down: “free propagation” instead of “freely propagation”.

Response: Thank you for your comments.

We are sorry for our negligence. The axis “x” in Figure 2 has been added, the “stimulated” has been replaced with “simulated”, and the “freely propagation” has been replaced modified to “free propagation”.

On p. 4 rows 2-4 from down I would propose: “..high Tm concentration doping was important to realize two-for-one cross-relaxation. Normally, materials doped with 4.0 at.% were used for…”

Response: Thank you for your comments.

We have changed the sentences to “We know that high Tm concentration doping was important to realize two-for-one cross-relaxation. Normally, materials doped with 4.0 at.% were used for laser operation”

  1. 5 row 5: it should be: “It is obvious that the multiple wavelength occurred and the peak wavelength blue-shifted rapidly when..”.

Response: Thank you for your comments.

We have changed the sentences to “It is obvious that the multiple wavelength occurred and the peak wavelength blue-shift rapidly when the OCs increased from 10% to 14%.”

  1. 5 row 2 from down: I would prefer “plane-parallel” instead of “plano-plano”.

Response: Thank you for your comments.

We have followed your suggestion.

  1. 6 row 3 from down (in Conclusion): should be: “…for the presented Tm:LuAG SCF laser.”.

Response: Thank you for your comments.

We have followed you suggestion and changed the sentence to “The results showed that the pure pump-guiding mode was not a good choice for the presented Tm:LuAG SCF laser.”